# SPIKING DECISION MAKING BOTTLENECK FOR OF-FLINE REINFORCEMENT LEARNING WITH SPIKING NEURAL NETWORKS

## ABSTRACT

Spiking Neural Networks (SNNs), with their event-driven low-power characteristics, provide key technological support in energy-constrained embodied intelligence applications, particularly in offline Reinforcement Learning (RL) tasks. However, offline RL, which relies solely on precollected data for policy training and cannot interact with the environment in real time, is limited by the inherent redundancy in offline data. This limits the model's ability to learn compact and generalizable representations, leading to degraded policy performance and reduced robustness. To address this issue, we propose the Spiking Decision Making Bottleneck (SDMB), a novel information compression framework designed for offline RL based on SNNs. The framework aims to guide the network in learning abstract and relevant trajectory representations for efficient policy learning. Specifically, it minimizes the mutual information between the input and latent representations, thereby suppressing input redundancy and promoting sparse, decision-relevant activations. To prevent over-compression and the consequent loss of critical behavioral information, SDMB further incorporates the principle of maximum entropy to ensure sufficient informational diversity is preserved during policy optimization. Experimental results on D4RL benchmark tasks validate the effectiveness of SDMB in extracting key spiking features in offline RL settings. Compared to both SNNs and Artificial Neural Networks (ANNs) methods,the performance of SDMB surpasses the state-of-the-art and achieves lower energy consumption, demonstrating dual advantages in energy efficiency and strategy generalization.

## 1 INTRODUCTION

Offline reinforcement learning (RL) algorithms hold great potential—transforming vast datasets into powerful decision engines. Efficient offline RL methods can extract the most useful strategies from precollected behavioral datasets Levine et al. (2020), driving the automation of decision-making in diverse fields such as healthcare, education, and robotics. However, despite significant progress in Artificial Neural Networks (ANNs)-based offline RL methods, such as conditional state modeling Chen et al. (2021); Janner et al. (2021) and value function approximation Kostrikov et al. (2021); Kumar et al. (2020), they often come with high computational and energy consumption costs, making them difficult to adopt in energy-constrained embodied intelligent applications.

As the third generation of neural networks Maass (1997), Spiking Neural Networks (SNNs) feature sparse, event-driven computations, where spikes are triggered only when a neuron's membrane potential reaches a threshold, making SNNs significantly more energy-efficient than ANNs and an ideal choice for embodied intelligent applications. Recent research has applied SNNs in Transformer-based architectures for tasks like image classification or object detection Deng et al. (2022); Fang et al. (2021); Guo et al. (2023); Kumar et al. (2022); Luo et al. (2024); Vaswani et al. (2017), yet offline reinforcement learning based on SNNs Peng et al. (2019); Tan et al. (2021); Huang et al. (2025) is still in its early exploratory stages.

Applying SNNs to offline RL presents unique challenges. One key issue is how to learn a more compact decision representation from offline data to enhance the robustness and generalization of the SNN model. During offline RL, precollected data inevitably contains noise generated by the

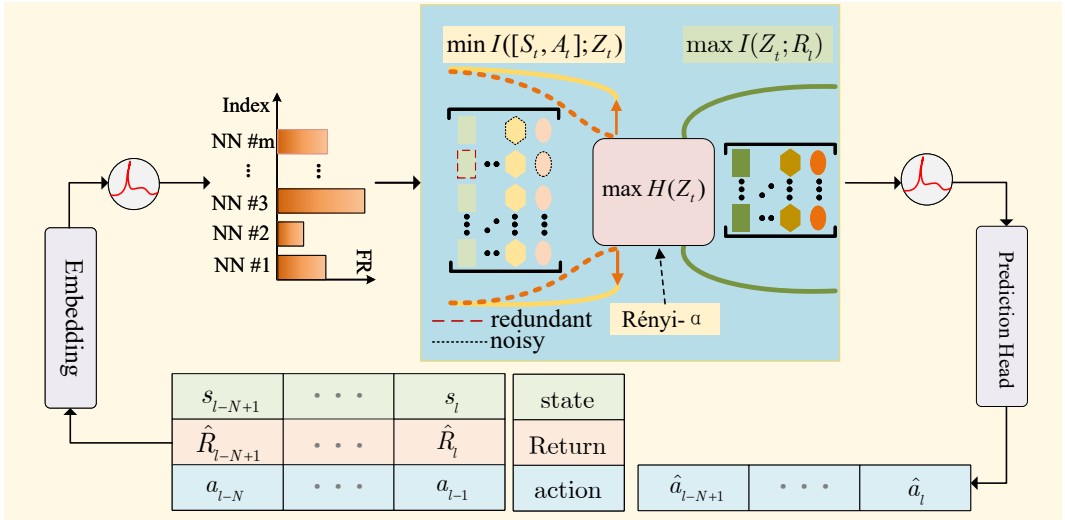

Figure 1: Spiking Decision Making Bottleneck Framework Diagram.This figure shows the overall flow of the SDMB strategy based on the SNNs reinforcement learning framework. The model takes the state $s$, reward $R_t$, and action sequence $a_t$ as inputs, and encodes them through a network layer to form the representation $Z_t$ in the SNNs layer. The spike firing rate (FR) of the neural layer is used to estimate the latent representation $Z_t$. In the model, the information compression is performed based on the Rényi-$\alpha$ entropy, which aims to minimize the mutual information between the input information and the latent representation $I([S_t, A_t]; Z_t)$ in order to remove noise. At the same time, to optimize the strategy's performance in terms of maximizing task-related information, the model maximizes the mutual information $I(Z_t; R_t)$ between the latent representation $Z_t$ and reward $R_t$. Additionally, SDMB introduces the max entropy term $\max H(Z_t)$, which helps mitigate the loss of information due to compression and improves the robustness of the latent representation. Finally, the latent representation is used to predict the action sequence $a_t$, which is output by the model. This framework combines the spiking neural network with reinforcement learning principles, providing insights into the interpretability and explainability of the model's decision-making process.

inherent randomness of the environment, sensor errors, uncertainty in behavior strategies, and data processing errors. On the other hand, due to the inherent similarity of state-action spaces, temporal correlations, limitations or preferences in behavior strategy exploration, and data collection biases, sampled data often contains redundant information. Since the model uses precollected data for policy training without real-time environmental interaction to correct data, reduce noise, and eliminate redundant information, existing SNN-based offline RL methods, relying solely on the neural network architecture, struggle to learn a more compact decision representation from offline data containing noise and redundancy, leading to limitations in model robustness and generalization.

To address this, in this study, we propose the Spiking Decision Making Bottleneck (SDMB), an SNNs-based information compression framework specifically tailored for offline reinforcement learning. As illustrated in Figure 1, in offline reinforcement learning, the trajectories generated by the behavior policy consist of states, actions, and rewards. Within the SDMB framework, we have constructed a Rényi-$\alpha$ entropy estimator using pulse frequency encoding. By adjusting the value of $\alpha$, we compress the mutual information between behavior trajectories and their latent representations, effectively suppressing the impact of noise and redundant information in the offline data on the encoding distribution. This allows the model to more easily learn compact representations from the behavior trajectories. Specifically, SDMB estimates the Rényi-$\alpha$ entropy of the state-action representations through statistical analysis of pulse frequencies in a low-dimensional space, and optimizes this entropy value to regulate the expressive power of the latent variables. This mechanism not only enhances the stability of training but also significantly improves the policy generalization performance in the presence of noise. Additionally, to enhance the bottleneck layer's ability to express useful task information, we introduce a maximum entropy regularization mechanism into the information compression objective, encouraging latent representations to retain rich informa-

tion while compressing redundancy. The maximum entropy alleviates the information loss caused by excessive compression, allowing SDMB to achieve an optimal balance between robustness and expressiveness.

We summarize our contributions as follows: 1) We propose the SDMB, which is the first application of the information bottleneck principle to offline reinforcement learning based on SNNs. This approach addresses the issue of limited model generalization due to noise and redundant information, providing a new possibility for learning compact and relevant trajectory representations from offline data. 2) To mitigate the information loss caused by excessive compression, we introduce the principle of maximum entropy, ensuring the preservation of sufficient informational diversity during the compression process. This allows for a balanced trade-off between the compactness and expressiveness of the feature representations. 3) Experimental results on the D4RL benchmark demonstrate that, under the same multi-task setup, our method outperforms the current state-of-the-art by 15.0%, and shows exceptional performance in noisy environments. Thanks to more efficient feature extraction, the proposed method also reduces theoretical energy consumption by 49.7%.

## 2 RELATED WORKS

### 2.1 OFFLINE REINFORCEMENT LEARNING

Offline RL is a variant of reinforcement learning where an agent learns from pre-collected historical data Levine et al. (2020), rather than interacting with the environment in real-time during training. Unlike traditional reinforcement learning methods, which typically require continuous interaction with the environment to receive feedback and improve, Offline RL relies on fixed datasets that include states, actions, and associated rewards. Additionally, Offline RL faces challenges related to data quality and coverage. Since historical data may contain noise or be incomplete, the agent may learn incorrect behaviors from suboptimal data. In offline reinforcement learning, early methods such as Behavior Cloning methods Torabi et al. (2018); Peng et al. (2019); Ashvin et al. (2020) treat offline data as supervised learning samples, directly learning the mapping from states to actions. However, such methods cannot surpass the behavior policy and are sensitive to suboptimal data. Subsequently, pessimistic value-based methods such as Conservative Q-Learning Kumar et al. (2020; 2022); Kostrikov et al. (2019) were introduced, which incorporate pessimistic bias in the Q-value estimates to suppress optimistic estimates for unseen state-action pairs. Recent Transformer-based methods Decision Transformer Chen et al. (2021); Trajectory Transformer Bucker et al. (2022) have transformed offline reinforcement learning into a Conditional Sequence Modeling problem Zhang et al. (2023), using Transformer to predict future actions, reducing the complexity of value function-guided optimization and enhancing model generalization. Other approaches leverage inverse reinforcement learning Kostrikov et al. (2021); IQ-Learn Garg et al. (2021) to infer the reward function and optimize the policy to maximize that reward.

### 2.2 SNNs METHODS FOR OFFLINE RL

Offline RL methods based on SNNs is a novel approach that integrates biologically inspired neural computing models with reinforcement learning techniques. In the Offline RL framework based on SNNs, SNNs are used to model the states, actions, and rewards from historical data. During the offline learning process, SNNs can gradually learn policies by extracting features from historical data without the need for real-time exploration. Currently, the vast majority of mainstream offline reinforcement learning methods are based on ANNs (include citations from the previous paragraph). Unlike traditional ANNs, Spiking Neural Networks, as the third generation of neural networks Maass (1997), achieve low-power operation through sparse, event-driven computation, making them particularly suitable for energy-constrained embodied learning scenarios. However, current SNNs-based offline reinforcement learning is still in its preliminary exploratory stage, with existing research mainly focusing on two paths: one approach first executes traditional offline RL strategies on ANNs architectures, then converts them into equivalent SNNs for performance inference deployment Peng et al. (2019); Tan et al. (2021). The other approach involves efficient trajectory modeling and policy learning through an SNNs model within the Decision Transformer (DT) framework, without introducing any additional value function-based regularization terms Huang et al. (2025). Although these methods can achieve low-power inference processes, the limited offline data, as well as inherent noise and redundant information in the data, constrain the generalization ability of SNNs

models. To address this issue, this study employs the information bottleneck principle to extract a more compact representation from offline data for decision-making, thereby enhancing the model's robustness and generalization.

# 3 PRELIMINARIES

## 3.1 SPIKING NEURAL NETWORKS

SNNs are a class of neural network models that mimic biological neural systems, characterized by information transmission through spikes rather than continuous signals, as seen in traditional neural networks. In SNNs, neurons generate spikes upon receiving external stimuli, and when the membrane potential reaches a certain threshold, a spike is emitted, which is then transmitted to other neurons through synaptic connections. Unlike traditional artificial neural networks, SNNs rely not only on the intensity of the input signal but also heavily on the timing and temporal differences of the signals. The spike emission of neurons is discrete and event-driven, making SNNs particularly efficient in processing temporal information and tasks driven by events, such as dynamic perception and time-series prediction. Additionally, SNNs are known for their low power consumption since computation only occurs when neurons emit spikes, making them especially suitable for embedded systems and neuromorphic computing platforms. Despite the significant advantages of SNNs in simulating biological neural systems, they still face substantial challenges in training and optimization compared to traditional deep learning networks. Due to their nonlinearity and temporal nature, the training process of SNNs is more complex. To improve the trainability of SNNs, researchers have proposed various methods based on spike coding, Spike-Timing-Dependent Plasticity (STDP), and other techniques, gradually advancing the application of SNNs in fields such as image processing, speech recognition, and robotic control.In the spiking neuron layer, we use the Leaky Integrate and Fire (LIF) model, which is a classical model for the membrane potential dynamics and spike generation mechanism of neurons. This model assumes that the membrane potential of a neuron increases over time due to incoming current and spikes once the membrane potential exceeds a threshold. The update equation for the membrane potential is given by:

$$U^t = H^{t-1} + I^t \tag{1}$$

where $H^{t-1}$ is the membrane potential at the previous time step and $I^t$ is the input current at the current time step. This equation describes the update process of the membrane potential.

$$S^t = Hea(U^t - U_{th}) \tag{2}$$

where $S^t$ is the condition for spiking at time step $t$, and $Hea(\cdot)$ is a Heaviside function. When $U^t$ exceeds the threshold $U_{th}$, the neuron fires, and $S^t = 1$; otherwise, $S^t = 0$.

$$H^t = U_{reset}S^t + \gamma U^t(1 - S^t) \tag{3}$$

If the neuron spikes, i.e., $S^t = 1$, the membrane potential is reset to $U_{reset}$; otherwise, the membrane potential decays based on the parameter $\gamma$.

## 3.2 OFFLINE RL

Reinforcement learning is often based on Markov decision processes (MDP), which provide a formal framework to describe reinforcement learning problems. An MDP is defined as a tuple $(S, A, P, R)$, where $S$ is the state space, $A$ is the action space, $P(s'|s, a)$ is the state transition probability, representing the probability of moving from state $s$ to state $s'$ when action $a$ is taken, and $R(s, a)$ is the reward function, representing the reward obtained by taking action $a$ in state $s$. In reinforcement learning, the agent's goal is to learn an optimal policy $\pi$ that maximizes the expected reward $E[R(\tau)]$, where $\tau$ represents a trajectory. The expected cumulative reward is defined as:

$$E[R(\tau)] = E\left[\sum_{t=0}^{T-1} \gamma^t r_t\right] \tag{4}$$

where $r_t$ is the reward obtained at time step $t$, $\gamma$ is the discount factor, and $T$ is the length of the trajectory. The offline reinforcement learning dataset $D$ is given by:

$$D = \{\, (s_0, a_0, r_0, s_1), (s_1, a_1, r_1, s_2), \ldots, \\ (s_{t-1}, a_{t-1}, r_{t-1}, s_t) \tag{5}$$

which contains the states and actions generated by the policy $\pi_B$.

## 4 METHODS

In this study, we propose an offline reinforcement learning strategy based on SNNs — the Spiking Decision-Making Bottleneck (SDMB) — aimed at enhancing the extraction of key spike features during the decision-making process. This method combines the principles of Information Bottleneck and maximum entropy regularization, utilizing Renyi-$\alpha$ entropy to compress redundant information, thereby enhancing the model's robustness and generalization capability by learning compact representations from offline data. The methodological framework is provided in the Appendix.

### 4.1 SDMB: MAIN IDEA AND OBJECTIVE

We first briefly introduce the Information Bottleneck (IB) principle. In supervised learning, the goal of the IB principle is to learn a compact representation $Z$ given input data $X$ and target variable $Y$, such that $Z$ preserves the information relevant to $Y$ while compressing irrelevant information as much as possible.

In offline reinforcement learning, since interaction with the environment is not possible, the training data often contains a large amount of redundant information or noise, which severely affects the robustness and generalization ability of SNNs models. Therefore, this paper proposes SDMB, which introduces the IB principle to build a latent representation $Z$ from noisy offline data that contains redundant information, enabling the extraction of decision-relevant features.

The objective of SDMB is to learn the most compact latent representation $Z$ from input variable $X$, while preserving the information relevant to the target variable $Y$. Considering both objectives and introducing a Lagrange multiplier $\beta$, the objective function can be written as: $\max I(Z; Y) - \beta I(Z; X)$, where $I(Z; Y)$ represents the mutual information between the latent representation $Z$ and the target variable $Y$, and $I(Z; X)$ is the mutual information between the latent representation $Z$ and the input variable $X$.

### 4.2 IMPLEMENTATION OF SDMB IN SNN-BASED OFFLINE RL

In the reinforcement learning task based on SNNs, where an MDP is defined by the tuple $(S, A, P, R)$. Therefore, we treat the state-action pair $(S_t, A_t)$ as the input variables to SDMB, and the reward variable $R_t$ as the target variable $Y$. The objective function of SDMB is as follows:

$$\max I(Z_t; Y) - \beta I\left([S_t, A_t]\,; Z_t\right) \tag{6}$$

In SNNs, the activity of neurons is often represented by the spike frequency. The firing rate of neurons in a given time window reflects the degree of response of the neurons to the input stimulus. In reinforcement learning and decision-making models, the important information regarding neuronal activity can also serve as a key optimization indicator for the model.

In our model, the calculation of firing rates is based on the number of spikes of each neuron at each time step. Specifically, we calculate the number of spikes from neurons at a given time step and normalize it by the time steps to obtain the average firing rate. This can be mathematically expressed as:

$$r_t = \frac{1}{T} \sum_{i=1}^{N} \delta(t_i - t) \tag{7}$$

where $r_t$ represents the firing rate within a time step, $T$ is the number of time steps, $N$ is the number of neurons in the time step, and $\delta(t_i - t)$ is an indicator function that specifies whether a spike occurred at time step $t$.

Through this method, we are able to evaluate the activity of neurons more accurately and further treat the processing as a base for optimization. The firing rate of neurons not only helps correct the rules of the neural network but can also enhance the decision-making process with efficient compression and information transmission.

To measure the information between the state-action pair $(S_t, A_t)$ and the encoded representation $r_t$, SDMB introduces the Rényi-$\alpha$ entropy as the compression indicator, which is defined as:

$$I_\alpha\left([S_t, A_t]; r_t\right) = \frac{1}{\alpha - 1} \log \mathbb{E}\left[\left(\frac{p(r_t|S_t, A_t)}{p(r_t)}\right)^\alpha\right] \tag{8}$$

where $\alpha$ is a parameter for controlling the entropy. By minimizing this entropy, we effectively control the compression degree of the input information. By varying the value of $\alpha$, the sensitivity of the entropy measure can be adjusted. When $\alpha$ approaches 1, the Rényi-$\alpha$ entropy degenerates to the classic Shannon entropy. When $\alpha$ is less than 1, the entropy measure becomes more sensitive to low-probability events, whereas when $\alpha$ is greater than 1, the sensitivity to high-probability events increases. Therefore, Rényi-$\alpha$ entropy provides a more flexible information measure than Shannon entropy, which can be adjusted based on specific application needs.This entropy measure has widespread applications in various fields, including data compression, image processing, and information transmission. In complex systems and machine learning, Rényi-$\alpha$ entropy is often used to assess the uncertainty of a system and to analyze the properties of high-dimensional and sparse data. By selecting an appropriate $\alpha$ value, Rényi-$\alpha$ entropy offers a more adaptive measure than Shannon entropy, especially when dealing with extreme data distributions, where it performs better.

Maximum Entropy is a method widely used in statistics, information theory, and physics to infer distributions under conditions of incomplete information. The basic idea is that, under certain conditions, we select the probability distribution that maximizes the entropy. The more uncertain a system is, the greater its entropy, and thus the maximum entropy principle suggests that the most appropriate probability distribution is one that maximizes uncertainty under the given constraints. Mathematically, maximum entropy is achieved by selecting the probability distribution that maximizes the entropy function, usually subject to known constraints such as expected values or marginal distributions.Maximum entropy methods have broad applications in various fields. In statistical inference, they are commonly used to infer the most appropriate probability distribution, especially when there is insufficient prior knowledge, thus avoiding biases in the data distribution. In natural language processing (NLP), maximum entropy models, such as the maximum entropy Markov model (MEMM), are widely used for classification tasks. In machine learning, maximum entropy methods are used to construct probabilistic models that solve optimization problems by maximizing entropy. Due to its lack of bias and flexibility, the maximum entropy principle has become a powerful tool, especially in situations with incomplete information.

To avoid over-compression leading to loss of relevant task-related information, SDMB also introduces the maximum entropy regularization term to maintain diversity and generalization ability:

$$\mathcal{L}_{\text{entropy}} = -\lambda H(r_t) \tag{9}$$

where the entropy $H(r_t)$ is given by:

$$H(r_t) = -\int p(r_t) \log p(r_t)\, dr_t \tag{10}$$

Finally, the loss function for SDMB is as follows:

$$\begin{aligned}
\mathcal{L}_{SDMB} =& I\left([S_t, A_t]; Z_t\right) - \beta I\left(Z_t; R_t\right) - \lambda H(Z_t) \\
=& \mathbb{E}_{p(S_t, A_t)}\left[\log p(Z_t | [S_t, A_t])\right] \\
& - \beta \mathbb{E}_{p(z_t)}\left[\log p(R_t | Z_t)\right] - \lambda H(Z_t)
\end{aligned} \tag{11}$$

Table 1: Results on MuJoCo.The strengthened digits denote the highest scores.

| MuJoCo Tasks | BC | CQL | DT | FCNet | SpikeGPT | SpikeBert | PSSA | SDMB |
|---|---|---|---|---|---|---|---|---|
| halfcheetah-m-e | 35.8 | 62.4 | 86.8±1.3 | **91.2±0.3** | 23.6±4.5 | 24.3±6.0 | 87.5±0.3 | 87.9±0.0 |
| walker2d-m-e | 6.4 | 98.7 | 108.1±0.2 | 108.8±0.1 | 22.6±4.8 | 92.5±22.4 | 108.7±0.1 | **109.0±0.0** |
| hopper-m-e | **111.9** | 111.0 | 107.6±1.8 | 110.5±0.5 | 32.7±5.4 | 84.1±8.8 | 91.5±0.2 | 98.7±0.3 |
| halfcheetah-m | 36.1 | **44.4** | 42.6±0.1 | 42.9±0.4 | 26.9±0.8 | 20.0±3.5 | 42.8±0.3 | 42.8±0.0 |
| walker2d-m | 6.6 | 79.2 | 74.0±1.4 | 75.2±0.5 | 16.4±10.2 | 22.9±10.4 | 75.2±1.4 | **78.3±0.1** |
| hopper-m | 29.0 | 58.0 | **67.6±1.0** | 57.8±6.0 | 25.1±6.4 | 31.4±4.9 | 58.3±4.3 | 63.5±28.4 |
| halfcheetah-m-r | 38.4 | **46.2** | 36.6±0.8 | 39.8±0.8 | 21.8±2.0 | 32.2±8.0 | 38.8±0.7 | 38.4±0.4 |
| walker2d-m-r | 11.3 | 26.7 | 66.6±3.0 | 63.5±7.5 | 16.7±3.3 | 21.2±6.4 | 71.7±3.6 | **73.0±0.9** |
| hopper-m-r | 11.8 | 48.6 | 82.7±7.0 | 85.8±1.7 | 51.5±7.1 | 30.1±8.6 | 86.3±0.3 | **87.3±24.1** |
| **MuJoCo mean** | 31.9 | 63.9 | 74.7 | 75.1 | 26.4 | 39.9 | 73.8 | **75.4** |

where $p(r_t|S_t, A_t)$ is the probability distribution of the pulse rate matrix given the state-action pair, and $p(R_t|r_t)$ is the probability of the reward given the encoded representation.

The loss function combines the information bottleneck framework, aiming for maximum compression and regularization of the input information. The goal is to effectively balance the accuracy of reward prediction and the diversity of variations in the model, thereby facilitating efficient learning through the use of off-line data.

In order to measure the model's generalization ability, we introduce the generalization bound. The following equation provides a measure for the maximum deviation between the true model and the trained model based on the training data:

$$\mathcal{L}(h) - \hat{\mathcal{L}}_s(h) \leq c \cdot \sqrt{\frac{2H_a\left(Z_t; [S_t, A_t]\right) + 2H(Z_t) + \log\frac{1}{\delta}}{n}} \tag{12}$$

In the equation above, the left-hand side represents the generalization error, i.e., the deviation between the model on actual data and the loss on the sample set. The right-hand side shows the bound on this error, including terms that account for the information and the diversity of the sample size $n$, ensuring that the generalization error is controlled.

In the previous analysis, we introduced the generalization error bound for the model's generalization ability. This theoretical result provides a measure for the model's ability to generalize to unseen data. Through this equation, we quantify the gap between the model's performance on training data and its performance on test data, and the potential for generalization error.

The key factors influencing generalization include model complexity, the diversity of the data, the representation of data, and the sample size. However, improvements in generalization are not only based on theoretical error bounds, but also closely related to the amount of data required for training. Thus, we proceed to derive a further bound for the generalization ability of the model, which helps ensure that the model's generalization ability remains within a given range of error. This is achieved by considering factors such as mutual information and the sample size.

By applying this framework, we can measure the impact of different factors (such as mutual information, and maximum entropy) on model generalization. To estimate this model's generalization ability, we use the following bound based on the loss function:

$$n \geq \frac{2L^2}{\epsilon^2}\left(\mathcal{L}_{SDMB}\left(Z_t; [S_t, A_t]\right) + \beta I\left(Z_t; R_t\right)\right.$$
$$\left. + (\lambda + 1)H(Z_t) \right) \tag{13}$$

In this case, $L$ is the Lipschitz constant in the hypothesis space, $\mathcal{L}_{SDMB}$ is the SDMB loss function, and $H(Z_t)$ is the entropy regularization term. $\beta$ and $\lambda$ are the hyperparameters used to balance the compression and prediction objectives.

Table 2: Results on Adroit.The strengthened digits denote the highest scores.

| Adroit Tasks | BC | CQL | DT | FCNet | SpikeGPT | SpikeBert | PSSA | SDMB |
|---|---|---|---|---|---|---|---|---|
| pen-e | 85.1 | 107.0 | 110.4±20.9 | 108.0±11.3 | 30.5±10.3 | 46.2±19.5 | 122.0±17.8 | **129.1±0.5** |
| door-e | 34.9 | 101.5 | 95.5±5.7 | 102.9±2.9 | 65.3±16.9 | 96.4±4.6 | 105.2±0.1 | **105.4±0.0** |
| hammer-e | 125.6 | 86.7 | 89.7±24.6 | 121.1±6.1 | 51.1±18.7 | 71.3±16.5 | 127.2±0.3 | **127.2±0.8** |
| relocate-e | 101.3 | 95.0 | 15.3±3.6 | 50.0±6.0 | 0.7±0.9 | 0.3±0.5 | **108.4±2.2** | 106.6±0.6 |
| pen-h | 34.4 | 37.5 | -0.2±1.8 | 57.7±11.1 | 29.8±11.7 | 20.0±16.7 | 75.7±25.1 | **89.8±5.6** |
| door-h | 0.5 | 9.9 | 0.1±0.0 | 0.4±0.5 | 0.1±0.0 | 0.2±0.0 | 0.2±0.0 | **21.9±29.3** |
| hammer-h | 1.5 | 4.4 | 0.3±0.0 | 1.2±0.0 | 0.3±0.0 | 0.3±0.0 | 0.2±0.0 | **2.7±1.3** |
| relocate-h | 0.0 | 0.2 | 0.2±0.2 | 0.0±0.0 | 0.1±0.0 | 0.0±0.0 | 0.0±0.0 | **0.5±0.1** |
| pen-c | 56.9 | 39.2 | 22.7±17.1 | 50.4±24.1 | 17.0±22.0 | 17.6±29.0 | 44.8±14.7 | **73.3±12.2** |
| door-c | -0.1 | 0.4 | 0.1±0.0 | -0.2±0.0 | 0.2±0.0 | 0.2±0.0 | 0.0±0.0 | **11.6±0.7** |
| hammer-c | 0.8 | 2.1 | 0.3±0.0 | 0.2±0.0 | 0.3±0.0 | 0.3±0.5 | 0.2±0.0 | **2.7±1.3** |
| relocate-c | -0.1 | -0.1 | -0.3±0.0 | -0.2±0.0 | **0.1±0.0** | 0.0±0.5 | -0.2±0.0 | 0.0±0.0 |
| **Adroit Mean** | 36.7 | 40.3 | 27.8 | 41.0 | 16.3 | 21.1 | 48.6 | **55.9** |

Table 3: Ablation Study

| Methods | Door-e | Door-h | Door-c | Average |
|---|---|---|---|---|
| SDMB | **105.5** | **28.2** | **12.1** | **48.6** |
| SDMB w.o. $L_{entropy}$ | 104.9 | 15.3 | 7.8 | 42.7 |
| PSSA | 105.2 | 0.2 | 0.0 | 35.1 |

Table 4: Energy consumption of various components of the model (μJ) on hopper-medium-replay dataset.

| | DT | PSSA | SDMB |
|---|---|---|---|
| Embedding | 0.9 | 0.9 | 0.9 |
| Self-Attention | 168.2 | 31.0 | 15.7 |
| MLP | 241.2 | 56.7 | 31.5 |
| Prediction Head | 0.2 | 0.2 | 0.2 |
| Total | 410.5 | 96.1 | 48.3 |

## 5 EXPERIMENTS

In this section, we evaluate the SDMB using the D4RL benchmark Fu et al. (2020) for a comprehensive assessment and report normalized scores following the protocol Fu et al. (2020) in , where a score of 100 represents expert-level performance, and a score of 0 reflects random agent performance. For specific experimental settings, refer to the Appendix. We compare SDMB with the following approaches: (1) Behavior Cloning Torabi et al. (2018). This method employs expert demonstrations to train the intelligent agent. (2) Conservative Learning Kumar et al. (2019). This method ensures stability and avoids overfitting through robust learning techniques, optimizing Q-values. (3) The Transformer method Chen et al. (2021), which uses sequence-to-sequence learning and applies Transformer architecture for sequence modeling in reinforcement learning. (4) The deep control network model Tan et al. (2024), which incorporates deep learning advantages such as backpropagation and frequency analysis for control tasks. It offers an effective control strategy, especially when dealing with complex sequences and periodic tasks. Additionally, there are models based on ANN, such as SpikeGPT Zhu et al. (2023), and SpikeBERT Lv et al. (2023), which combine the benefits of ANNs and SNNs models.

Finally, we discuss the current SOTA method, DSFormer Huang et al. (2025), which designs a DT framework to build a model using position self-attention (PSSA) for capturing dependencies between the sequence order and positional features necessary for learning tasks.

### 5.1 THE MAIN RESULTS

**Results on MuJoCo** We evaluate the performance of all methods on HalfCheetah, Hopper, and Walker2d, as shown in Table 1, which presents the experimental results demonstrating the superiority of SDMB. In this table, 'm' stands for 'medium'; 'm-r' stands for 'medium-replay'; 'm-e' stands for 'mediumexpert', which combines both the medium and expert policies. SDMB ranks second in all subtasks and achieves the highest average score. This is because MuJoCo tasks typically have a high-dimensional action space, with large amounts of noise and unreliable information, while SDMB can reduce irrelevant information from the action space, resulting in more efficient action

selection. Further analysis is provided in the ablation study. The poor performance of other SNNs methods is due to the shortcut connection in SpikeBERT that introduces integer values, disrupting residual propagation and affecting spike characteristics, while SpikeGPT, designed for NLP, suffers from the limited data in offline RL.

**Results on Adroit** To evaluate the model's performance in handling precise manipulation and complex interactions, we tested all methods on four tasks in the Adroit dataset: Pen, Door, Hammer, and Relocate. The experimental results are shown in Table 2, where 'e' stands for 'expert', 'h' stands for 'human', and 'c' stands for 'cloned', representing the respective policies. The results show that SDMB significantly outperforms other ANN and SNNs methods, surpassing human policy strategies and behavior cloning strategies. This is because, compared to expert policy strategies, the high-dimensional action space generated by human policy strategies contains more noisy information, and the SDMB framework effectively reduces irrelevant information, aiding the model in capturing the key behaviors of action strategies. Further analysis is provided in the ablation study.

## 5.2 ABLATION

To investigate the roles of various components of the loss function, we compare SDMB and the SDMB w.o. the largest entropy constraint $L_{\text{entropy}}$ on the Adroit dataset's 'Door' task. This helps to avoid random behavior due to the entropy constraint.In the experiment, we use the same random seed (seed=2025). The results, shown in Table 3, indicate that adding the largest entropy constraint improves the accuracy of each sub-task. This means that the largest entropy constraint prevents overfitting and ensures that SDMB retains a balanced amount of information during the strategy optimization process.

## 5.3 POWER CONSUMPTION

We begin by analyzing the computational power consumption of the SDMB model in SNNs and comparing it with the performance of both ANNs and SNNs models. First, we calculate the FLOPs based on the spike firing rates of individual neurons in each layer and then aggregate these to compute the total computational cost of the entire model. The final FLOP and power consumption results are presented in Table 4. The results show that the SDMB model exhibits lower computational power consumption compared to the ANN-based DT model. When compared to the SNN-based PSSA, SDMB demonstrates a reduction in computational power consumption by 49.7%. This indicates that SASIB effectively compresses input data, thereby improving the energy efficiency of the model.

## 6 CONCLUSION

This research proposes a strategy called SDMB, which aims to address the challenge of offline learning in SNNs. By constructing an estimator based on Rényi- entropy for mutual information, SDMB reduces the bottleneck caused by noise and irrelevant information in the compressed data, allowing the model to efficiently perform strategy learning. Meanwhile, SDMB introduces the largest possible filter to prevent excessive compression and the resulting loss of key information, ensuring that the strategy optimization process retains the most relevant information, which is essential for maintaining the model's dynamic equilibrium. Experimental results show that SDMB significantly improves model performance compared to previous methods, while also significantly reducing computational complexity. This work highlights the new potential applications of SNNs in areas requiring high computational efficiency, demonstrating a new approach for extracting relevant features efficiently.

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

# A APPENDIX

## A.1 ETHICS STATEMENT

This paper focuses on the study of Spiking Neural Networks (SNNs) with the aim of improving the compactness of offline data extraction and optimizing the generalization and robustness of SNNs models. We ensure that all research adheres to the principles outlined in the ICLR Code of Ethics and does not involve any violations of ethical guidelines.

## A.2 REPRODUCIBILITY STATEMENT

We have made efforts to ensure the reproducibility of the results presented in this paper. In the appendix, we provide details of the experimental setup, as well as the corresponding pseudocode for the methods. Additionally, we will release the full code of this work on GitHub once the paper is publicly published, allowing the community to reproduce the results.

## A.3 LARGE LANGUAGE MODELS USAGE STATEMENT

The present study was conducted without the use of any Large Language Models (LLMs) or LLM-based tools throughout its entire process, including conceptualization, experimental design, data processing, result analysis, and manuscript preparation. All text composition, figure generation, and analytical work were independently performed by the authors, relying solely on conventional academic methodologies and human expertise. The findings presented herein represent the original contributions of the research team, without reliance on generative AI systems.

## A.4 PSEUDOCODE

This pseudocode describes the training process of the SDMB in offline reinforcement learning. SDMB is an information compression framework aimed at optimizing the learning process of neural networks by minimizing the redundancy between input data and latent representations, thereby

improving the efficiency and generalization capability of policy learning. The training process primarily consists of two stages: the forward pass and the backward pass, each of which involves updating the network weights and microscopic parameters.

---

**Algorithm 1** Training Process of SDMB

---

1: **Input:** Event data $(X)$; label $(Y)$
2: **Output:** Updated network weights $(W)$; updated microscopic parameters $(\lambda', \beta')$
3: **Parameters:** SNN neuron parameters $(\lambda', \beta')$; SASIB hyperparameters $(\lambda, \beta)$; batch size $(B)$;

4:      training epochs $(TE)$; learning rate $(r)$; network depth $(H)$; neuronal membrane potential $(V)$ and spike output $(O)$
5: **Forward Pass:**
6: 1: Initialize model and optimizer
7: 2: Preprocess input data to obtain batch data
8: **for** $i \leftarrow 1$ **to** $TE$ **do**
9:     Fetch a mini-batch $B$
10:     **for** $h \leftarrow 1$ **to** $H - 1$ **do**
11:         Calculate $V_h$ and $O_h$
12:     **end for**
13:     Calculate spike rates and trajectory representation
14:     Calculate $V_h$ and $O_h$
15:     Calculate the loss $L_{\text{loss}} = \text{SASIB}(I, L_{\text{flow}}(Y, \hat{Y}), \beta, H(T))$
16: **end for**
17: **Backward Pass:**
18: **for** $h \leftarrow H$ **to** $1$ **do**
19:     **if** $h \neq H$ **then**
20:         Update the network weights
21:         $\Delta W_h = \sum \frac{\partial L_{\text{loss}}}{\partial W_h}$
22:         $W_h \leftarrow W_h - r\Delta W_h$
23:         Update the microscopic parameters
24:         $\Delta \lambda_h = \sum \frac{\partial L_{\text{loss}}}{\partial \lambda_h}$
25:         $\lambda_h \leftarrow \lambda_h - r\Delta \lambda_h$
26:         $\Delta \beta_h = \sum \frac{\partial L_{\text{loss}}}{\partial \beta_h}$
27:         $\beta_h \leftarrow \beta_h - r\Delta \beta_h$
28:     **else**
29:         Update the network weights
30:         $\Delta W_h = \sum \frac{\partial L_{\text{loss}}}{\partial W_h}$
31:         $W_h \leftarrow W_h - r\Delta W_h$
32:     **end if**
33: **end for**
34: **Helper Functions:**
35: Calculate Spiking Rate:
36: spiking rate $= \frac{1}{T} \sum_{i=1}^{N} \delta(t_i - t)$
37: Calculate Maximum Entropy:
38: $H(Z) = -\sum \rho(z) \log p(z)$
39: Mutual Information Calculation:
40: $I(Z_t, Y) = \max \left( I(Z_t, Y) - \beta \cdot I(S_t, A_t, Z_t) \right)$

---

First, the training process begins by initializing the model and optimizer. The model initialization phase includes initializing the network weights $(W)$ and microscopic parameters $(\lambda', \beta')$, as well as setting hyperparameters such as the learning rate $(r)$ and network depth $(H)$. During each training epoch, the forward pass begins by fetching a mini-batch of data, followed by computations for each layer's neurons to obtain membrane potentials $(V_h)$ and spike outputs $(O_h)$. Next, the spiking rates for each neuron layer are computed, and the SDMB framework is used to calculate the mutual information loss $(I_{\text{loss}})$ and the maximum entropy loss (max_entropy_loss). The SDMB loss function

combines mutual information minimization and maximum entropy regularization, yielding the total loss ($L_{loss}$), which provides the basis for parameter updates during the backward pass.

During the backward pass, the network weights are updated starting from the final layer. If the current layer is not the final layer ($h \neq H$), the network weights ($W_h$) and microscopic parameters ($\lambda_h$, $\beta_h$) for that layer are updated. The network weights are updated using gradient descent by calculating the gradient of the loss function with respect to the weights. Similarly, the microscopic parameters are updated using the same method. Once the backward pass is complete, the network weights and microscopic parameters are updated.

At the end of each training epoch, the model continues to update the network weights until the pre-determined number of training epochs ($TE$) is reached. This iterative process enables the network to progressively optimize its learning, ensuring better generalization and low energy consumption in offline reinforcement learning environments.

Additionally, the pseudocode includes helper functions to compute the spiking rate, maximum entropy, and mutual information. These functions are crucial to the optimization process, ensuring that the network can efficiently handle sparse data while maintaining information diversity.

Overall, this pseudocode combines SNNs with the SDMB framework, utilizing maximum entropy and mutual information minimization for policy learning. It provides an effective solution to address the issue of data redundancy in offline reinforcement learning.

## A.5 THE SELECTION OF DATASETS

### A.5.1 CHARACTERISTICS OF MUJOCO DATASETS

The MuJoCo dataset typically refers to collections of trajectory data generated within the MuJoCo physics engine simulation environment (e.g., scenarios widely used in benchmark datasets like D4RL). Its core characteristics stem from MuJoCo's high-fidelity physics engine and specialized task design, making it an ideal testbed for Reinforcement Learning and Information Bottleneck tasks. Key attributes include high-dimensional continuous state and action spaces, precise and efficient physical simulation, rich contact dynamics modeling, diverse task configurations and difficulty gradients, large-scale standardized trajectory data, and inherent potential for low-dimensional structured representations. Firstly, the MuJoCo physics engine provides exceptionally realistic and computationally efficient simulations of rigid bodies, soft bodies, and contact dynamics. It accurately models complex physical phenomena like gravity, joint friction, collision responses, and actuator dynamics. This high-fidelity physics forms the bedrock of MuJoCo datasets, ensuring that generated trajectories (containing state sequences, action sequences, reward signals, etc.) reflect core challenges from real-world robotic or biomechanical control tasks (e.g., balance maintenance, goal-directed locomotion, disturbance rejection). Consequently, RL agents trained on this data learn strategies with potential physical realism and transferability value. Secondly, MuJoCo tasks typically involve high-dimensional, continuous state spaces (e.g., joint angles, angular velocities, end-effector positions, object poses) and high-dimensional, continuous action spaces (e.g., direct joint torque or target angle outputs). This continuous high-dimensional nature closely mirrors the essence of real-world robotic control problems (e.g., humanoid walking, robotic arm manipulation), avoiding simplifications inherent in discrete spaces. It compels RL algorithms to effectively handle high-dimensional input-output mapping, the exploration-exploitation trade-off, and continuous policy fine-tuning, providing a robust test of an algorithm's generalization capability, stability, and sample efficiency in complex scenarios.

This continuous high-dimensionality makes MuJoCo a natural application ground for the Information Bottleneck principle. High-dimensional observations (like raw joint sensor streams) inevitably contain streams of information redundant or potentially detrimental for predicting future states or maximizing cumulative rewards. The IB objective precisely requires learning a compressed, intermediate representation retaining only task-relevant information, filtering out noise or task-irrelevant physical details. The structured constraints inherent in MuJoCo's physics (e.g., kinematic chain constraints, chain reactions of contact forces) provide a clear environment for verifying potential optimal solutions for this information compression and purification. Physical systems exhibit strong spatiotemporal regularities, suggesting the existence of compact, task-relevant representations. MuJoCo datasets allow explicitly quantifying how IB techniques extract these representations from

high-dimensional sensory data by minimizing the mutual information between the representation and the raw input while preserving information about future states or rewards. Thirdly, MuJoCo tasks are highly modular and diverse (e.g., different robot morphologies like Ant, HalfCheetah, Hopper, Walker, Humanoid, or object interaction tasks like door-opening, hammer-wielding). Standard datasets (e.g., D4RL) provide large-scale, standardized offline trajectory data generated under various policy levels, including expert demonstrations, sub-optimal policies, random exploration, and partially observable perturbations, covering a broad spectrum from near-optimal behavior to random exploration. This richness and standardization are crucial for RL:

- It supports Offline RL algorithm training and evaluation. Researchers can test how effectively algorithms learn optimal or near-optimal policies solely from fixed, static datasets while avoiding dangerous or erroneous extrapolation errors (*out-of-distribution* actions).
- Task diversity ensures comprehensive algorithm evaluation, mitigating the risk of overfitting to a single environment and demonstrating broader applicability.

For IB research, this multi-task, multi-policy dataset introduces crucial variables: task objective (different reward signals), behavior policy (data source), and state representation (raw or encoded). This allows researchers to precisely quantify and optimize the mutual information between the learned state representation and critical variables (e.g., the behavior policy or the task reward) under the constraint of maximizing task performance. This validates whether applying the IB framework within RL pipelines enhances robustness, generalization, and policy interpretability by forcing the representation to ignore irrelevant distractors. The ability to compare representations learned from different policy distributions (expert vs. random) is particularly valuable for disentangling task-relevant information from artifacts of data collection. Finally, while providing high physical accuracy, MuJoCo simulations maintain relatively high computational efficiency, enabling rapid generation of vast amounts of interaction data. This is crucial for training deep RL models and optimizing IB representations, both of which often require massive amounts of experience.

Simultaneously, the underlying physical laws exhibit explicit structural characteristics:

- **Local Smoothness:** State space dynamics are locally smooth under physical constraints. Small changes in state/action lead to predictable changes in next state/reward.
- **Sparsity and Temporal Dependency:** Contact dynamics exhibit sparsity (few significant contacts at any moment) and strong instantaneous dependencies.
- **Low-Dimensional Control:** Complex motion patterns often emerge from low-dimensional latent variables (e.g., gait phase).

These characteristics provide inherent justification for learning low-dimensional representations (the core goal of IB). An agent can theoretically leverage these structures to discard redundant high-dimensional information (e.g., tiny joint tremors) and focus on critical features like gait phase, contact state flags, or goal-related vectors. This pursuit of a Minimal Sufficient Statistic is a quintessential application of the IB principle. MuJoCo datasets inherently encapsulate this structured information, making them an ideal proving ground for verifying that IB methods improve RL agents' robustness, generalization, and interpretability.

In summary, MuJoCo datasets, characterized by their physical fidelity, high-dimensional continuous spaces, task complexity, data richness, and inherent structured information properties, provide an unparalleled platform. They serve as a core pillar for testing and advancing methodologies at the intersection of Reinforcement Learning and Information Bottleneck theory. The dataset's realism enables transfer studies, while its well-defined simulation structure offers control and precise measurement essential for theoretical analysis of information flow and representation learning.

### A.5.2 CHARACTERISTICS OF THE ADROIT DATASET AND COMPARISON WITH MUJOCO BENCHMARKS

The Adroit dataset represents a specialized benchmark constructed on the MuJoCo physics engine, focusing exclusively on high-dexterity manipulation tasks using a sophisticated 24-degree-of-freedom (DoF) anthropomorphic robotic hand model. This dataset preserves MuJoCo's foundational characteristics of high-fidelity physical simulation and continuous high-dimensional state/action spaces, where states typically encompass joint angles, velocities, end-effector positions, object

poses, and even fingertip tactile forces, while actions involve precise control of joint torques or target angles. However, Adroit introduces distinctive characteristics that significantly diverge from conventional MuJoCo locomotion tasks (such as Ant, Hopper, or Walker environments that primarily involve balance and movement), establishing it as a more advanced testing ground for cutting-edge algorithms.

The defining characteristics of Adroit stem primarily from its exceptional task complexity and precision requirements. Unlike the relatively macroscopic mobility or balance objectives in MuJoCo, Adroit demands agents to coordinate multiple finger joints with extreme precision to execute sequential operations involving tool usage, fine grasping, force modulation (such as holding a pen without slipping or crushing it), and complex object interactions. This anthropomorphic manipulation paradigm naturally produces extreme reward sparsity, where success signals (such as fully hammering a nail, completely opening a door lock, or completing a pen twirl) typically emerge only at the conclusion of lengthy and precise action sequences, rendering the vast majority of exploratory trajectories reward-free and creating substantial exploration bottlenecks. Furthermore, Adroit's sensory inputs exhibit increased complexity and multimodality, augmenting basic proprioception (joint data) with tactile feedback and often exteroceptive information, resulting in higher-dimensional, more heterogeneous input streams. Additionally, Adroit tasks are inherently long-horizon and multi-phase, with successful outcomes typically requiring the correct sequential completion of interdependent subtasks (e.g., approaching a door handle, grasping it, rotating it, and finally pulling the door open), imposing substantial demands on long-term planning and policy robustness. Finally, given the exceptional difficulty of these tasks, the Adroit dataset heavily relies on expert demonstration data, supplemented by suboptimal or random policy data, creating a crucial yet exceptionally challenging foundation for offline reinforcement learning.

When compared against the broader landscape of MuJoCo benchmarks, both environments share core physics engines and continuous high-dimensional spaces. The crucial distinctions emerge in: task category (Adroit: dexterous object manipulation versus MuJoCo: locomotion/balance), agent morphological complexity (Adroit: anthropomorphic multi-fingered hand versus MuJoCo: relatively simpler limbs or torsos), sensory modality richness (Adroit: proprioception + tactile + exteroception ¿ MuJoCo: primarily proprioception), reward sparsity characteristics (Adroit: extreme sparsity versus MuJoCo: typically denser rewards, often including velocity incentives), temporal structure (Adroit: long-horizon, multi-phase versus MuJoCo: shorter, cyclic patterns), exploration difficulty (Adroit: exceptionally high versus MuJoCo: moderate), and data composition (Adroit: heavy dependence on expert demonstrations versus MuJoCo: broader data sources including RL agent interactions). The distinctive characteristics of the Adroit dataset establish it as an exemplary platform for evaluating and advancing reinforcement learning methodologies, directly targeting critical limitations in contemporary algorithms. The combination of extreme reward sparsity and high-precision manipulation requirements creates an exceptionally challenging exploration problem where traditional random exploration-based RL methods prove largely ineffective. This necessity compels the development of sophisticated exploration mechanisms such as intrinsic motivation or curiosity-driven approaches, or alternatively, demands highly efficient utilization of expert prior knowledge through imitation learning or demonstration-enhanced RL. Concurrently, the long-horizon, multi-phase nature of tasks complicates long-term credit assignment, requiring temporal difference methods and policy gradient algorithms to precisely attribute eventual success to early critical decisions despite sparse and delayed reward signals. The complex multimodal state space coupled with high-dimensional continuous action spaces continuously tests the representational capacity and optimization stability of function approximators like deep neural networks. Moreover, the Adroit dataset, particularly through its inclusion of expert demonstrations and mixed-quality trajectories, positions it as a foundational benchmark for offline reinforcement learning. Algorithms must learn high-performance policies exclusively from static datasets, avoiding catastrophic extrapolation errors due to distributional shift while potentially striving to surpass demonstrator performance. The inherent complexity and phased structure of the tasks also naturally connects to research in hierarchical reinforcement learning and transfer learning, creating opportunities to explore skill reuse or compositional sub-policy architectures within these demanding manipulation contexts.

The Adroit dataset presents a uniquely valuable and challenging environment for applications of Information Bottleneck theory. Its high-dimensional, heterogeneous, multimodal state space, integrating joint kinematics, tactile sensations, and object pose information, inherently contains substantial redundancy and irrelevant data. The core IB objective—learning compressed representations that

minimize information from the raw input while maximizing relevance to task objectives like future states or cumulative reward—becomes particularly crucial here. IB provides a principled framework to guide models in filtering out noise, such as minor joint vibrations or irrelevant contact point signals, and instead focus on extracting features vital to the task, such as indicators of stable grasps, effective tool contact points, or precursors to slippage, which are essential for efficient decision-making in dexterous manipulation. Facing the challenge of extreme reward sparsity, the IB principle can steer the learning of representations that maximize predictive information about future task progression or eventual reward. This capability helps agents identify subtle perceptual cues (like minute object movement trends or specific force patterns) potentially predictive of success during the early stages of extended action sequences lacking immediate feedback. The inherent over-parameterization of control in high-dimensional dexterous tasks suggests significant scope for information compression. IB's pursuit of minimal sufficient statistics aids in discovering the most parsimonious and robust control representations, enhancing policy generalization and interpretability. Furthermore, the common inclusion of multi-source data in Adroit offers unique potential for IB to disentangle task-relevant information from strategy-specific idiosyncrasies. This enables learning representations that encode only the core physical constraints and object properties essential to the task, independent of a particular demonstrator's habitual motions, which is critical for generalization beyond demonstrations and subsequent performance improvement. Finally, IB methods hold promise for effectively processing and fusing multimodal inputs with disparate noise profiles and temporal characteristics, extracting complementary information necessary to support robust manipulation policies.

Building upon the MuJoCo physics simulation foundation, the Adroit dataset, through its specialized focus on anthropomorphic robotic hand dexterity tasks, introduces critical challenges including extreme reward sparsity, complex multimodal perception, and long-horizon multi-phase decision-making. These characteristics markedly differentiate it from general MuJoCo locomotion benchmarks, pushing the boundaries of reinforcement learning difficulties—particularly concerning efficient exploration and long-term credit assignment—to unprecedented levels, while simultaneously establishing high standards for offline reinforcement learning research. For information bottleneck investigations, Adroit's high-dimensional heterogeneous state space and intrinsic information redundancy create a fertile ground for learning and validating minimally sufficient representations capable of efficiently filtering noise, extracting task-critical features, processing multimodal inputs, and decoupling policy styles. Consequently, Adroit serves not only as the preeminent benchmark for assessing RL algorithms tackling real-world-grade manipulation challenges but also stands as an ideal proving ground for developing and evaluating Information Bottleneck theory in enhancing agent perception, decision robustness, and interpretability within embodied artificial intelligence systems striving for human-level dexterity.

### A.6 THEORETICAL ENERGY EVALUATION

The main indicator for evaluating the energy consumption of neuromorphic chips is the average energy consumption through spike-driven transmissions, which is a key factor in the energy consumption of the entire processing system and plays an important role in the overall energy consumption. For hardware-related theoretical analysis, we view the whole system as a single spiking operation (SOP), and treat the energy consumption per SOP as a constant. The models of SDMB and the control group are shown in Table 4. The energy consumption model for this approach can be expressed as:

$$E = C_E \cdot \#SOP = C_E \cdot \sum_i s_i c_i \qquad (14)$$

where $C_E$ represents the energy consumption of each spiking operation, and $\#SOP = \sum_i s_i c_i$ represents the total number of spiking operations. For each spiking neuron $i$, $s_i$ represents the spike count of the neuron, while $c_i$ represents the synaptic connection count of the neuron. It is worth noting that this energy consumption model may not be suitable for all hardware architectures. We believe it is particularly suitable for high-density and sparse architectures.

| Operation | DT | PSSA | SDMB |
|-----------|-----|------|------|
| Embedding | $D_{\text{src}}D_{\text{N}}$ | $D_{\text{src}}D_{\text{N}}$ | $D_{\text{src}}D_{\text{N}}$ |
| $Q, K, V$ | $3D^2N$ | $3D^2NR_m$ | $3D^2NR_m$ |
| $f(Q,K,V)$ | $(2D+3)N^2$ | $TDN^2\hat{R}$ | $TSDN^2R_m$ |
| Attn Linear | $D^2N$ | $D^2NR_m$ | $D^2NR_m$ |
| MLP Linear1 | $4D^2N$ | $4TD^2NR_m$ | $4TD^2NR_m$ |
| MLP Linear2 | $4D^2N$ | $4TD^2NR_m$ | $4TD^2NR_m$ |
| Prediction Head | $D_{\text{tgt}}D_{\text{N}}$ | $D_{\text{tgt}}D_{\text{N}}$ | $D_{\text{tgt}}D_{\text{N}}$ |

Table 5: The FLOPs of various operations. $R_m$ and $\hat{R}$ refer to the sum of spike firing rates across different spiking matrices.

## A.7 THE PROOF OF THE FORMULA

### A.7.1 THE PROOF OF GENERALIZATION ERROR BOUND

I suggest first defining the model's generalization error and training error, and then considering how to reduce generalization error by increasing the sample size. Generalization error is typically expressed as:

$$L_{\text{gen}} = L_{\text{train}} + \mathcal{O}\left(\frac{1}{\sqrt{n}}\right) \tag{15}$$

where $L_{\text{train}}$ is the training error and $n$ is the sample size. This bound indicates that as the sample size increases, the generalization error tends to decrease. The second part of the formula indicates the difference between the error on the training data set and the error on the true data set. We derive an upper bound of the generalization error by comparing the error between different data sets. Assuming we have training error and the model's generalization ability:

$$L(h) - \hat{L}_S(h) \leq c \times \left(2H \cdot [Z_t|S_t, A_t] + 2\lambda H(Z_t) + \log\frac{1}{\delta}\right) \tag{16}$$

By comparing the training error and generalization error between data sets, we derive the relationship between them. Finally, we aim to reduce the generalization error by increasing the sample size $n$, thereby improving the model's generalization ability.

### A.7.2 COMPLEXITY BOUND DERIVATION

We derive the complexity bound formula in the context of model training and generalization. The complexity bound is influenced by the model's Lipschitz constant, the loss function, and the sample size. The formula we are deriving is as follows: The generalization error is typically expressed as:

$$L_{\text{gen}} = L_{\text{train}} + \mathcal{O}\left(\frac{1}{\sqrt{n}}\right) \tag{17}$$

where $L_{\text{train}}$ is the training error, and $n$ is the sample size. This bound indicates that as the sample size increases, the generalization error will decrease. The second part of the formula represents the difference between the error on the training data set and the true data set. We derive the upper bound of generalization error by comparing errors across different data sets. Assuming we have training error and the model's generalization ability, the error bound is expressed as:

$$L(h) - \hat{L}_S(h) \leq c \times \left(2H \cdot [Z_t|S_t, A_t] + 2\lambda H(Z_t) + \log\frac{1}{\delta}\right) \tag{18}$$

The formula for the complexity bound is derived as follows:

$$n \geq \frac{2L^2}{\epsilon^2}\left(L_{\text{SDMB}} + \beta I(Z_t, R_t) + (\lambda+1)H(Z_t)\right) \tag{19}$$

where:$L$ is the Lipschitz constant of the model. $L_{\text{SDMB}}$ is the SDMB loss function. $\beta$ and $\lambda$ are regularization parameters. $I(Z_t, R_t)$ is the information gain between state $Z_t$ and reward $R_t$.$H(Z_t)$ represents the entropy of $Z_t$. $n$ is the sample size.$\epsilon$ is the error tolerance. The formula gives the minimum number of samples $n$ required to achieve a specified complexity level in terms of generalization error. It shows that, as $n$ increases, the model can handle more complex data, allowing it to refine its performance.

