# OpenReview forum: "Spiking Decision Making Bottleneck for Offline Reinforcement Learning With Spiking Neural Networks"
_ICLR.cc/2026/Conference — ICLR 2026 Conference Withdrawn Submission_

### Official Review · Reviewer_AVnu · 2025-10-25

**Soundness:** 2
**Presentation:** 2
**Contribution:** 2
**Rating:** 4
**Confidence:** 3

**Summary:**

This paper proposes the Spiking Decision Making Bottleneck (SDMB), a novel offline reinforcement learning (RL) framework that incorporates the Information Bottleneck (IB) principle and maximum entropy regularization into Spiking Neural Networks (SNNs). The goal is to learn compact, noise-robust representations from pre-collected offline data, addressing the generalization challenges in SNN-based RL. The method is evaluated on the D4RL benchmark, including MuJoCo and Adroit tasks, and compared against both SNN and ANN baselines.

**Strengths:**

1. Apply the Information Bottleneck principle to SNN-based offline RL is an interesting idea.
2. Energy Efficiency: The paper highlights the energy advantages of SNNs, showing that SDMB reduces theoretical energy consumption by nearly 50% compared to other SNN methods, which is significant for neuromorphic deployment.
3. The paper is well-organized, with a logical progression from introduction to methodology, experiments, and conclusions, making it accessible to readers

**Weaknesses:**

1. The paper lacks a systematic analysis of how different α values affect learning dynamics or representation quality. The choice of α appears empirical without theoretical backing.
2. The training objective combines multiple loss terms (IB, entropy, prediction), each with its own hyperparameters. The sensitivity of performance to these hyperparameters is not thoroughly explored, raising concerns about reproducibility and practical deployment.
3. The paper does not compare against recent strong offline RL methods such as IQL, IFQL, FQL and so on. These methods may offer competitive performance even under the energy constraints of SNNs, and their omission limits the scope of the empirical evaluation.

**Questions:**

1. Since the authors have not provided code, could the reproducibility of the work be demonstrated by visualizing the training curves？
2. The authors did not test their method on the AntMaze environment. Does this imply that the approach is unsuitable for sparse-reward scenarios?
3. There is a lack of evidence that Rényi entropy is irreplaceable in this method.

---

### Official Review · Reviewer_qycx · 2025-10-26

**Soundness:** 4
**Presentation:** 3
**Contribution:** 3
**Rating:** 8
**Confidence:** 5

**Summary:**

This paper proposes a novel information bottleneck framework for offline reinforcement learning based on Spiking Neural Networks (SNNs). The method guides the network to learn abstract and relevant trajectory representations by minimizing the mutual information between the input and latent representations, and extends the bottleneck capacity through the incorporation of the maximum entropy principle. The approach is theoretically grounded, deriving generalization bounds and sample complexity bounds, and demonstrates performance improvements on the MuJoCo and Adroit datasets while reducing energy consumption.

**Strengths:**

1. The method is supported by rigorous theoretical analysis, including the derived generalization error and sample complexity bounds, which strengthen the foundation of the proposed information bottleneck framework.

2. The method effectively enhances the model's ability to learn compact and generalizable representations from offline datasets, surpassing traditional methods for offline reinforcement learning.

3. The integration of the maximum entropy principle enhances the flexibility and representational capacity of the bottleneck.

4. Experimental results demonstrate improved accuracy and reduced energy consumption compared to existing methods.

**Weaknesses:**

1. Although the paper demonstrates the performance improvement of the proposed method through experiments, a deeper exploration of the impact of key parameter settings on the results would further strengthen its practical foundation.

2. The paper could also benefit from a more thorough theoretical justification for the specific choices made in the method, such as the introduction of Renyi-α entropy and maximum entropy regularization, and why these are particularly advantageous for offline reinforcement learning tasks.

3. While the paper mentions that the method reduces energy consumption, further elaboration on how these reductions are achieved would be beneficial.

**Questions:**

1. The paper introduces Renyi-α entropy as a key component in the SDMB framework for compressing redundant information. Could the authors clarify why Renyi-α entropy is specifically chosen over traditional Shannon entropy or other entropy measures for this task?

2. The paper introduces a maximum entropy regularization term to prevent excessive information loss during compression. Could the authors provide a more detailed theoretical justification for why maximum entropy regularization is particularly effective in this context? Are there alternative regularization methods that could achieve similar results?

---

### Official Review · Reviewer_vjbe · 2025-10-27

**Soundness:** 1
**Presentation:** 1
**Contribution:** 1
**Rating:** 0
**Confidence:** 4

**Summary:**

Offline Reinforcement Learning aims to learn a policy for acting in an environment from a fixed set of observed trajectories. The authors hypothesize that offline-RL can benefit from information compression, for example through an information bottleneck approach, and demonstrate this principle as their SDMB method for spiking neurons. Compared to Sota for spiking neural networks, they improve on two benchmark suites and claim half the estimated energy consumption compared.

**Strengths:**

Strenghts:
The work extensively benchmarks their proposed approach against standard benchmark suites.

**Weaknesses:**

- the paper is poorly written and structured, the introduction in particular reads like imprecise and over-the-top AI-written text.
- the problem definitions are imprecise. What exactly is learned in offline reinforcement learning, what is the actual goal, what is meant by "the inherent redundancy in offline data"? All this needs precise descriptions.
- many definitions are omitted. What is \pi_b? What is "N" in (7), why choose a population rate code like that? How does one obtain the probabilities in (8)?
- the related work is outdated and incomplete, citing work offline RL only up to 2023, for example missing the SOTA for ANNs (quick search shows that at least Omori et al 2025 exceeds all scores), this work then only slightly improves over PSSA for SNNs. Note that Omori improved Sota for ANNs substantially using a simple filtered behavioral cloning approach.
- the use of spiking neurons seem irrelevant: the model is an integrate-and-fire neuron (NOT LIF!), and only rates from a population of spiking neurons are used. This should mean that the same approach should work for a simple non-spiking RNN structure as well.
- while loss-functions are defined, the learning itself is not defined. How is the network optimised to best perform? This links to the lack of clear defintions.
- the energy consumption estimate is only shown for hopper-medium-replay. Is this a representative choice, or
- the energy consumption model only considers SOPs, and ignores the cost of multiplications vs additions in different architectures. Given the claimed improvement (50%), this matters for the claims.
- figure 1 is unclear. What do the different shapes mean?
- Table 1 and Table 2 need explicit definitions of what the different methods are. I can infer that "BC" refers to Behavioral Cloning, but the others are always obvious.
- SDMB lacks citations to the main idea of Information Bottleneck as it is formulated here.
-

**Questions:**

See weaknesses.

---

### Official Review · Reviewer_j5gA · 2025-10-31

**Soundness:** 3
**Presentation:** 3
**Contribution:** 3
**Rating:** 4
**Confidence:** 4

**Summary:**

This paper proposes the Spiking Decision Making Bottleneck (SDMB), an information compression framework for offline reinforcement learning based on Spiking Neural Networks (SNNs). The method applies Information Bottleneck principles using Rényi-α entropy to learn compact representations from noisy offline data, while incorporating maximum entropy regularization to prevent over-compression. Experiments on D4RL benchmark tasks demonstrate improved performance and reduced energy consumption.

**Strengths:**

Novel Application: The paper presents the first application of Information Bottleneck principles to SNN-based offline RL, which is a meaningful contribution to an underexplored research area.

Well-Motivated Problem: The motivation for addressing noise and redundancy in offline data for SNNs is clear and relevant, particularly for energy-constrained embodied AI applications.

Comprehensive Experiments: The evaluation covers both MuJoCo and Adroit benchmarks, demonstrating consistent improvements (15.0% over SOTA) and significant energy reduction (49.7%).

Theoretical Grounding: The paper provides theoretical analysis including generalization bounds and complexity bounds, connecting the practical method to information theory.

**Weaknesses:**

Equation 8: The Rényi-α entropy formulation I_α([S_t, A_t]; r_t) is unclear. How is this computed in practice? The expectation over p(r_t|S_t, A_t)/p(r_t) needs detailed derivation.

Firing Rate as Representation: The jump from spike firing rates (Eq. 7) to treating r_t as the latent representation Z_t is not well justified. Why is firing rate sufficient to capture all relevant decision information?

Loss Function (Eq. 11): The final loss function contains log p(Z_t|[S_t, A_t]) and log p(R_t|Z_t), but these conditional probabilities are never explicitly modeled or parameterized. How are these distributions estimated?

Several grammatical errors (e.g., "the largest entropy constraint" should be "maximum entropy")
Inconsistent notation (Z_t vs r_t as latent representation)

**Questions:**

See weaknesses

---

### Note · Authors · 2025-11-28

I have read and agree with the venue's withdrawal policy on behalf of myself and my co-authors.